# Impact of Microbiota and Metabolites on Intestinal Integrity and Inflammation in Severe Obesity

**DOI:** 10.3390/ph17070918

**Published:** 2024-07-10

**Authors:** Emma Custers, Debby Vreeken, Frank Schuren, Tim J. van den Broek, Lieke van Dongen, Bram Geenen, Ivo de Blaauw, Maximilian Wiesmann, Eric J. Hazebroek, Robert Kleemann, Amanda J. Kiliaan

**Affiliations:** 1Department of Medical Imaging, Anatomy, Radboud University Medical Center, Radboud Alzheimer Center, Donders Institute for Brain Cognition and Behaviour, Center for Medical Neuroscience, 6500 HB Nijmegen, The Netherlands; emma.custers@radboudumc.nl (E.C.); bram.geenen@radboudumc.nl (B.G.); maximilian.wiesmann@radboudumc.nl (M.W.); 2Department of Bariatric Surgery, Vitalys, Rijnstate Hospital, 6815 AD Arnhem, The Netherlands; ehazebroek@rijnstate.nl; 3Department of Metabolic Health Research, The Netherlands Organization for Applied Scientific Research (TNO), 2333 BE Leiden, The Netherlands; frank.schuren@tno.nl (F.S.); robert.kleemann@tno.nl (R.K.); 4Division of Pediatric Surgery, Department of Surgery, Radboudumc-Amalia Children’s Hospital, 6525 GA Nijmegen, The Netherlands; ivo.deblaauw@radboudumc.nl; 5Division of Human Nutrition and Health, Wageningen University and Research, 6708 WE Wageningen, The Netherlands

**Keywords:** obesity, microbiota, intestinal inflammation, intestinal integrity

## Abstract

Obesity is a multifactorial disease associated with low-grade inflammation. The gut is thought to be involved in obesity-related inflammation, as it is continuously exposed to antigens from food, microbiota and metabolites. However, the exact underlying mechanisms are still unknown. Therefore, we examined the relation between gut pathology, microbiota, its metabolites and cytokines in adults with severe obesity. Individuals eligible for bariatric surgery were included. Fecal and plasma samples were collected at surgery timepoint, to assess microbiota and metabolite composition. Jejunal biopsies were collected during surgery and stained for cytotoxic T cells, macrophages, mast cells and tight junction component zonula occludens-1. Based on these stainings, the cohort was divided into four groups: high versus low intestinal inflammation and high versus low intestinal integrity. We found no significant differences in microbiota diversity between groups, nor for individual bacterial species. No significant differences in metabolites were observed between the intestinal inflammatory groups. However, some metabolites and cytokines differed between the intestinal integrity groups. Higher plasma levels of interleukin-8 and tauro-chenodeoxycholic acid were found, whereas isovaleric acid and acetic acid were lower in the high intestinal integrity group. As the results were very subtle, we suggest that our cohort shows very early and minor intestinal pathology.

## 1. Introduction

The increased prevalence of obesity has become a huge health problem worldwide. Obesity is a multifactorial disease associated with low-grade inflammation [1]. However, underlying mechanisms driving obesity-associated inflammation are poorly understood. Nonetheless, the gut is thought to be involved as it is continuously exposed to antigens from food components, microbiota, its metabolites and cytokines.

It has been established that the gut microbiota plays a role in the development and function of the immune, metabolic and nervous system [2]. Therefore, the decreased diversity of the gut microbiota often found in obesity, known as dysbiosis, may lead to several pathological disorders [3]. Moreover, in dysbiosis, the abundance of harmful gut bacteria is often increased, which may increase cytokine release and cause mucosal inflammation and damage to intestinal epithelial cells [3,4]. Further evidence for the development of inflammation by the gut microbiota comes from studies on inflammatory bowel disease (IBD). Species of the *Enterobacteriaceae* phylum, in particular *Proteobacteria*, are often found to be highly abundant in IBD [5], whereas common families of *Firmicutes* such as *Ruminococcaceae* and *Lachnospiraceae* are typically reduced in IBD patients [6]. Additionally, *Faecalibacterium prausnitzii* and *Ruminococcaceae* are reduced in IBD [7], suggesting a role of the microbiota in intestinal inflammation. As intestinal inflammation is associated with reduced intestinal barrier function [3,4,8], the gut microbiota could induce intestinal permeability through intestinal inflammation. Moreover, in an in vitro model of an intestinal barrier monolayer, it was demonstrated that a combination of several gut bacteria, including *Lactobacillus Rhamnosus*, *Bifidobacterium Lactis* and *Bifidobacterium Longum,* increased the expression of tight junction proteins, including zonulin-1 and -2, occludin and claudin-1, indicating a role of the microbiota in intestinal integrity [9].

Gut microorganisms are able to ferment nutrients into absorbable forms and are critical for the fermentation of indigestible carbohydrates into short-chain fatty acids (SCFAs) [10]. These SCFAs constitute an important energy source for intestinal epithelial cells and are thought to strengthen the intestinal barrier [11]. They have a protective effect against bacterial pathogens and promote anti-inflammatory and immunomodulatory effects [10,11,12]. For example, *F*. *prausnitzii* produces butyrate [13], which is the main energy source for colonocytes, therewith enhancing the regeneration of colonocytes and maintaining intestinal integrity and reducing inflammatory intestinal diseases [14]. Furthermore, butyrate induces the development of regulatory T cells to promote intestinal mucosal immune tolerance and preserves the balance between Th17 and regulatory T-cell development in order to minimize intestinal inflammation [13,15]. Moreover, in the liver, cholesterol is synthesized into primary bile acids, which are converted into secondary bile acids by the gut microbiota [16]. In obesity, an altered primary-to-secondary bile acid pool might be involved in low-grade intestinal inflammation, due to the pro-inflammatory properties of primary bile acids on intestinal epithelial cells [16]. Contrarily, some secondary bile acids can exhibit anti-inflammatory effects and are therefore considered to regulate metabolic-inflammatory host homeostasis [16]. Finally, increased endogenous production of lipopolysaccharides (LPSs) by Gram-negative bacteria can also induce inflammation and lower intestinal barrier integrity [17]. These observations highlight the importance of the gut microbiota and gut-derived metabolites in the maintenance of intestinal health.

Although levels of several plasma inflammatory markers have been associated with adiposity in obesity, increasing evidence reveals that intestinal barrier dysfunction may instigate and/or exacerbate chronic low-grade inflammation in obesity [18]. However, the exact underlying mechanisms are poorly understood. The present study therefore examines the relation between gut pathology, fecal microbiota, plasma microbiota-derived metabolites and cytokines in adults with severe obesity enrolled in the BARICO study (BAriatric surgery Rijnstate and Radboudumc neuroImaging and Cognition in Obesity) [19]. Identifying associations between gut microbiota and gut pathology in patients with obesity may provide useful insights into the relationship between obesity, intestinal inflammation and intestinal barrier dysfunction. This may eventually contribute to the development of treatments for obesity-related intestinal alterations.

## 2. Results

### 2.1. Descriptive Statistics

In total, 100 individuals, of which 82% were female, were included in the analysis. Participant characteristics are listed in Table 1. The mean age of the participants was 46.5 ± 5.73 years with a mean body mass index (BMI) of 41.61 ± 3.92 kg/m^2^, a waist circumference (WC) of 124.75 ± 11.14 cm, systolic blood pressure of 136.72 ± 16.30 mm Hg and diastolic blood pressure of 84.78 ± 7.96 mm Hg. In our cohort, 14 individuals had diabetes and 71 were hypertensive. Moreover, 6 individuals were smokers and 40 consumed alcohol. Some patients had diseases related to the gastrointestinal tract. In particular, five patients suffered from inflammatory bowel syndrome and one patient from ulcerative colitis (Table 1). No differences were found in patient characteristics between the low and high intestinal inflammation group. In the high intestinal integrity group, the number of females (*p* = 0.007) and body length (*p* = 0.013) were significantly higher compared to the low intestinal integrity group.

### 2.2. Intestinal Inflammation and Integrity

The jejunum of the participants in our cohort had a mean cell count of 1040.16 ± 285.17 cytotoxic T cells/mm^2^, 628.36 ± 308.72 macrophages/mm^2^, 103.18 ± 66.19 mast cells/mm^2^ and a mean zonula occludens-1 (ZO-1) intensity of 76.34 ± 6.48. Besides the higher abundance of macrophages (*p* < 0.001) and mast cells (*p* < 0.001), no significant differences were found between the low and high intestinal inflammation groups (Table 1, Figure 1). In the high intestinal integrity group, the ZO-1 intensity (*p* < 0.001) and number of macrophages (*p* = 0.002) were significantly higher compared to the low intestinal integrity group (Table 1, Figure 2). 

### 2.3. Intestinal Microbiota

No significant differences were found in microbiota alpha and beta diversity between the groups with high and low intestinal inflammation (*p* = 0.069, *p* = 0.905), nor between the high and low intestinal integrity groups (*p* = 0.942, *p* = 0.694, Appendix A). Finally, counts of individual gut bacteria did not differ between groups (Appendix A).

### 2.4. Metabolites

To assess differences in metabolites between the intestinal inflammation and integrity groups, sex was included as a covariate. No significant differences were found in metabolite concentrations between intestinal inflammation groups (Appendix A). The significantly different metabolites between the high and low intestinal integrity groups are depicted in Figure 3 and Appendix A. Levels of tauro-chenodeoxycholic acid (TCDC) (*p* = 0.019) were significantly higher, whereas levels of iso valeric acid (*p* = 0.003) and acetic acid (*p* = 0.043) were significantly lower in the high intestinal integrity group. No significant differences were found for cholic acid (CA), chenodeoxycholic acid (CDC), deoxycholic acid (DCA), glycocholic acid (GCA), glycochenodeoxycholic acid (GCDC), glycodeoxycholic acid (GDC), glycolitocholic acid-3-sulphate (GLC-3S), tauro-deoxycholic acid (TDC), ursodeoxycholic acid (UDC), propionic acid, butyric acid, iso butyric acid and methyl butyric acid between the high and low intestinal integrity group (Appendix A).

### 2.5. Cytokines

Sex was included as a covariate to study cytokine concentrations in the intestinal inflammation and integrity groups. No significant differences were found in cytokine concentrations between intestinal inflammation groups (Appendix A). The significantly different metabolites between the high and low intestinal integrity groups are depicted in Figure 3 and Appendix A. Levels of IL-8 (*p* = 0.011) were significantly higher in the high intestinal integrity group compared to the low intestinal integrity group. No significant differences were found for high-sensitivity (hs) C-reactive protein (CRP), serum amyloid A (SAA), haptoglobin, LPS binding protein (LBP), tumor necrosis factor alpha (TNFα), interleukin-1β (IL-1β), IL-6, IL-4 and IL-10 between the high and low intestinal integrity group (Appendix A).

## 3. Discussion

In this study, the impact of microbiota, its metabolites and cytokines on intestinal barrier function and inflammation was investigated by comparing groups with high and low jejunal inflammation as well as groups with high and low jejunal integrity based on histological criteria. No significant differences were found in alpha and beta microbiota diversity between groups exhibiting high versus low intestinal inflammation and high versus low gut integrity. However, some differences in cytokines and metabolites were found. In particular, IL-8 and TCDC levels were higher, whereas iso valeric acid and acetic acid levels were lower in the high intestinal integrity group. The high and low intestinal inflammation groups did not differ in cytokine nor metabolite concentrations. 

Monteiro-Sepulveda and co-workers demonstrated that classical cytotoxic T cells and macrophages have a higher abundancy in individuals with obesity compared to lean individuals [20], whereas mast cell density was similar in obesity and lean controls. More specifically, they found 1550 cytotoxic T cells/mm^2^, 150 macrophages/mm^2^ and 100 mast cells/mm^2^ in patients with obesity, compared to 1000 cytotoxic T cells/mm^2^, 100 macrophages/mm^2^ and 100 mast cells/mm^2^ in lean controls. In our study, we found a similar cytotoxic T cell count (1040 cells/mm^2^) compared to the lean control group in the study of Monteiro-Sepulveda and colleagues (1000 cells/mm^2^). However, the macrophage cell density (628 cells/mm^2^) was higher in our study, compared to the cohort with obesity (150 cells/mm^2^) and lean controls (100 cells/mm^2^) in the study of Monteiro-Sepulveda et al. Another study showed approximately 900 cytotoxic T cells/mm^2^ and approximately 1100 macrophages/mm^2^ in the lamina propria of the colon from patients with Crohn’s disease compared to 200 cytotoxic T cells/mm^2^ and 550 macrophages/mm^2^ in healthy controls [21]. In addition, it has been found that macrophages were increased in the lamina propria of the duodenum in patients with Crohn’s disease (110 cells/mm^2^) compared to controls (50 cell/mm^2^) [22]. In accordance, Bottois et al. showed similar proportions of cytotoxic T cells in the ileum of patients with Crohn’s disease and in control ileum tissue [23]. Although, previous studies used different techniques and different regions of the intestine to analyze immune cell densities, these results suggest that patients with (severe) obesity participating in our study exhibited a certain degree of inflammation in the gut mucosa, although the pathology does not appear to be as severe as in Crohn’s disease. 

Animal and human studies reveal that obesity is associated with changes in gut microbiota diversity and composition. In particular, these studies describe a lower abundance of *Bacteroidetes* but a higher abundance of *Firmicutes* in the microbiota of obese individuals [24]. Also, LPS levels in obesity were higher in the colonic epithelia and were associated with increased inflammation and impaired colonic epithelial barrier function, as evidenced by increased pro-inflammatory cytokines, decreased mucus layer thickness and increased intestinal permeability [24]. Nonetheless, we found no significant differences in gut microbiota diversity between the high and low intestinal inflammation and integrity groups. The lack of differences in gut microbiota between the intestinal inflammation and permeability groups may suggest that the gut microbiota in our cohort is still healthy and functionally intact. Another possibility may be that the gut microbiota is slightly disrupted in all groups. However, no control group was included in this study, making it difficult to draw conclusions regarding quality of the gut microbiota.

It has been reported that microbiota-derived metabolites can influence inflammatory processes in the intestine as well as in other organs, as they are able to enter the systemic circulation through intestinal epithelial cells [25]. However, in our study, no differences in metabolites were found between the high and low intestinal inflammation groups. This is consistent with the lack of differences in microbiota diversity between the groups, suggesting that the microbiota composition was comparable and the production of metabolites such as secondary bile acids and SCFAs was not affected. Of note, our low intestinal inflammation group exhibited higher counts of macrophages compared to previously reported intestinal immune cell densities in individuals with obesity [20]. Therefore, it is also possible that no differences in metabolites were detected because the low inflammation group was de facto already slightly inflamed, masking any differences. Another possibility could be that the microbiota production of SCFAs and bile acids is not yet affected by the observed differences in intestinal inflammation in obesity. Nonetheless, we suggest that our cohort represents marginally elevated intestinal inflammation, but conclusions should be drawn carefully because of the lack of a control group in this study. 

Previously, it has been demonstrated that a higher jejunal permeability in patients with obesity correlated with systemic and intestinal inflammation [8]. Moreover, reduced expression levels of tight junction proteins such as ZO-1 and E-cadherin were found in the colon of patients with IBD [18]. In addition, an increase in pro-inflammatory cytokines and chemokines, such as IL-6, TNFα and IL-8, was found in patients with IBD [18]. In our study, the high and low intestinal inflammation groups did not show variations in ZO-1 intensity, reflecting no differences in intestinal integrity. This could suggest that the jejunal integrity is still preserved in our cohort, despite the relatively high immune cell count. 

We detected higher IL-8 and TCDC plasma levels, as well as lower levels of iso valeric acid and acetic acid, in individuals with a high intestinal integrity. IL-8 is known to be a chemoattractant of neutrophils which form an important line of defense against bacterial pathogens [26], suggesting that higher levels of IL-8 could maintain a higher intestinal integrity. However, the mean concentration of IL-8 in both the low and high permeability groups is still within the normal range (<62 pg/mL) [27], which may suggest that pathology may be limited in these groups. In an organoid-derived epithelial monolayer culture from a patient with ulcerative colitis, it was shown that acetate stimulation could prevent alterations of the monolayer integrity upon inflammation, suggesting that acetate has barrier-protective properties [28]. Moreover, in human colonic Caco-2 cells, it was shown that acetate maintains gut integrity by increasing cell survival [29]. In our study however, the high intestinal integrity group showed lower acetic acid levels. Therefore, we compared the acetic acid concentration of our cohort to the normal range of acetic acid in healthy people (3.6 µg/mL) [30] and found little differences. Furthermore, iso valeric acid and TCDC are negatively associated with Crohn’s disease and inflammation [18,31,32], while in our high intestinal integrity group, these levels where higher compared to the low intestinal integrity group. Nonetheless, the mean TCDC concentration of both the low and high inflammation group are similar to those of healthy individuals (0.057 µmol/L) [33]. Thus, the pathology in our cohort seems far less severe compared to Crohn’s disease and ulcerative colitis, which could explain the observed subtle results. These findings reveal that our cohort probably represents early intestinal pathology, explaining the subtle differences in intestinal integrity between groups and making it difficult to draw conclusions regarding the link between intestinal barrier function and microbiota-derived metabolites.

### 3.1. Limitations 

This study has some limitations. First, we did not include a control group, making it difficult to interpret the results and draw conclusions on inflammatory load in individuals with obesity compared to healthy subjects. Nonetheless, it is difficult to obtain jejunum biopsies from healthy individuals, therefore, we separated our cohort into low and high intestinal inflammation and intestinal integrity groups. Therewith, the low inflammation group and high intestinal integrity group functioned as reference groups. Furthermore, we compared our data with previous studies which investigated intestinal inflammation in patients with obesity, Crohn’s disease as well as healthy controls. However, analyzing methods may have differed which could influence results, making it difficult to compare our dataset with others. Second, our cohort did not have an equal sex distribution, as most of the participants, intrinsic to the type of surgical procedure [34], were female. It is important to consider this unequal sex distribution, as the epidemiology and pathophysiology of obesity and its accompanying metabolic disorders may differ between sexes [35] and potentially influence intestinal inflammation and integrity. Nonetheless, the sex distribution of our cohort represents the general BS population [34]. 

### 3.2. Conclusions 

In conclusion, this cohort showed the infiltration of immune cells in the jejunum of individuals with severe obesity. However, no differences in microbiota, its metabolites, cytokines and ZO-1 intensity were detected between patients with obesity who have high versus low intestinal inflammation, suggesting that there is no link between intestinal inflammation, the microbiota, its metabolites, cytokines and intestinal integrity at this stage of obesity. Furthermore, no differences in microbiota diversity were found between the intestinal integrity groups. However, IL-8, TCDC, iso valeric acid and acetic acid differed between individuals with high and low intestinal integrity. We therefore suggest that in this stage of obesity, the jejunal mucosa was still preserved with relatively little pathology despite the relatively high cell counts of immune cells. Furthermore, the difference in ZO-1 expression between groups may be too subtle to investigate the link between intestinal barrier function and microbiota-derived metabolites and cytokines. Future studies should concentrate on intestinal inflammation and permeability in lean individuals as well as individuals with obesity. It would also be interesting to assess differences between patients with obesity who are metabolically healthy and those with at least one comorbidity, as immune cell density differs between these conditions [32]. Finally, mechanistic rationales of how gut microbiota and metabolites can alter intestinal inflammation and integrity should be incorporated. Such studies will provide further insights into the relationship between intestinal microbiota, intestinal inflammation and intestinal barrier dysfunction in obesity and may contribute to the development of treatments for obesity-related intestinal alterations.

## 4. Materials and Methods

### 4.1. Description of Study Population

In this study, data of the BARICO study [19] were analyzed. Between September 2018 and December 2020, 156 patients, screened and found eligible for bariatric surgery (BS) based on the Fried guidelines [36], were recruited at the Rijnstate Hospital (Arnhem, The Netherlands). All subjects underwent Roux-en-Y gastric bypass (RYGB) surgery and were aged between 35 and 55 years at the time of recruitment. Neurological or severe psychiatric illness, pregnancy and treatment with any antibiotics, probiotics, or prebiotics 3 months before inclusion were exclusion criteria. 

Participants underwent a medical evaluation four weeks before surgery. Fecal samples were collected one week before surgery and plasma was collected on the day of surgery. Moreover, jejunal biopsies were collected during RYGB surgery. For the current study, only participants from which both plasma and fecal samples and a jejunal biopsy were present were included. Of the 156 participants, 56 were excluded, leaving 100 participants eligible for analysis (Figure 4).

### 4.2. Standard Protocol Approvals, Registration and Patient Consents

The study was approved by the Medical Ethics Committee CMO region Arnhem—Nijmegen (NL63493.091.17) and by the local institutional ethics committee. The study was conducted in accordance with the Declaration of Helsinki ‘Ethical Principles for Medical Research Involving Human Subjects’ and in accordance with the guidelines for Good Clinical Practice (CPMP/ICH/135/95). All participants provided written informed consent. The study was prospectively registered in the Netherlands Trial Registry (NL7090: https://www.clinicaltrialregister.nl/nl/trial/28949 (accessed on 10 May 2024)).

### 4.3. Medical Examination

Anthropometric measurements included body weight, BMI, WC and blood pressure. BMI was calculated as body weight divided by height in meters squared. Blood pressure was measured in sitting position. Hypertension was defined as the use of antihypertensive drugs and/or blood pressure ≥130/80 mm Hg. Diabetes type 2 mellitus was defined by the use of oral antidiabetic or insulin medication and/or fasting plasma glucose (FPG) of ≥7.0 mmol/L and HbA1c ≥ 48 mmol/mol, and prediabetes was defined by FPG ≥ 5.5 and <7.0 mmol/L and HbA1c ≥ 39 and <48 mmol/mol. Smoking was defined by smoking more than 1 cigarette per day.

### 4.4. Biochemical Analysis in Plasma

Fasting blood samples were collected and stored at −80 °C on the day of surgery. The analysis of bile acids (CA, CDC, DCA, GCA, GCDC, GDC, GLC-3S, TCDC, TDC and UDC; µmol/L) and SCFAs (acetic acid, propionic acid, butyric acid, iso butyric acid, methyl butyric acid and isovaleric acid; µg/mL) was performed at Triskelion (Utrecht, the Netherlands) by ultra-performance liquid chromatography (Ultimate 3000 UPLC; Thermo Scientific, Waltham, MA, USA). The UPLC unit was coupled to a high-resolution mass spectrometer (HR-MS; Q-Exactive mass spectrometer equipped with an electro-spray ionization probe; Thermo Scientific, Waltham, MA, USA) and EDTA plasma samples (50 µL aliquots) were analyzed as described in detail previously [37]. The performance of the measurement unit was controlled throughout the analysis, i.e., by injecting 50 µL aliquots from a large (7 mL) reference EDTA plasma pool at regular intervals. Plasma biomarkers of inflammation were analyzed using ELISA following established protocols and optimized conditions [38,39]. More specifically, the following ELISAs purchased from R&D Systems (Abingdon, UK) were used: hs-CRP, µg/mL (D1707); SAA, µg/mL (D3019); haptoglobin (µg/mL) (D8465-05); LBP, µg/mL (DY870-05); TNF-α, pg/mL; IL-1β, pg/mL; IL-6, pg/mL; IL-4, pg/mL; IL-10, pg/mL; and IL-8 pg/mL, which were determined with multiplex technology using the SP-X™ imaging system (Quanterix, Billerica, MA, USA).

### 4.5. DNA Isolation

Feces samples were collected and stored at −80 °C one week before surgery. For DNA isolation, fecal samples were thawed on ice and lysed by bead beating (mini-BeadBeater-24, Biospec Products, Bartesville, OK, USA) for 2 min at 2800 oscillations per minute in the presence of 800 µL of lysis buffer (Dneasy 96 Powersoil Pro QIAcube HT kit, Qiagen, Hilden, Germany) and 500 µL zirconium beads (0.1 mm; Biospec products, Bartlesville, OK, USA). DNA was extracted using the Dneasy 96 Powersoil Pro QIAcube HT kit (Qiagen, Hilden, Germany) in accordance with the manufacturer’s recommendations. DNA quality was assessed by routine gel electrophoresis as well as by capillary electrophoresis on a Fragment Analyzer (Advanced Analytical, Heidelberg, Germany).

### 4.6. Amplicon Sequencing

Microbiota composition was analyzed using 16S rDNA amplicon sequencing. The V4 hypervariable region was targeted. An amount of 100 pg of DNA was amplified as described elsewhere [40] with the exception that 30 cycles were used instead of 35, applying F515/R806 primers [41]. Primers included Illumina adapters and a unique 8-nt sample index sequence key [40]. The amplicon libraries were pooled in equimolar amounts and purified using the QIAquick Gel Extraction Kit (QIAGEN, Hilden, Germany). Amplicon quality and size were analyzed on a Fragment Analyzer (Advanced Analytical Technologies, Inc., Heidelberg, Germany). Paired-end sequencing of amplicons (approximately 400 base pairs) was conducted on the Illumina MiSeq platform (Illumina, Eindhoven, The Netherlands).

Sequence pre-processing, analysis and classification was performed using the DADA2 (version 1.14) software package in R [42]. Chimeric sequences were identified and removed using the ‘removeBimeraDenovo’ function from DADA2. The non-chimeric amplicon sequence variants (ASVs) were taxonomically classified using the assignTaxonomy function against the Silva nr 138 reference database. Taxonomic classification was performed up to the genus level.

### 4.7. Jejunal Histopathology

During RYGB surgery, jejunum samples were obtained and immediately fixed in 4% formaldehyde for 24–48 h. Samples were dehydrated overnight (Automatic Tissue Processor ASP300S, Leica Biosystems, Amsterdam, the Netherlands) and embedded in paraffin. Then, 5 µm thick cross-sections were mounted on Superfrost glass slides for subsequent immunohistochemical analysis. 

### 4.8. Immunohistochemistry

Sections were stained for hematoxylin-eosin (HE) following standard histology protocols. (Immuno-)histochemistry was performed on adjacent sections for macrophages (Monoclonal Mouse Anti-Human CD68, Clone KP1, DAKO, RRID: AB 2314148), cytotoxic T cells (Monoclonal Mouse Anti-Human CD8, Clone C8/144B, DAKO, RRID: AB 3073940) and mast cells (Monoclonal Mouse Anti-Human Mast Cell Tryptase, Clone AA1, DAKO, RRID: AB 2206478) at the Pathology department (Radboud University Medical Center, Nijmegen, The Netherlands). 

Sections were first deparaffinized in xylene, rinsed through graded ethanol series and finally in demi water. Thereafter, CD8 and CD68 epitope retrieval was performed for 10 min in EnVision FLEX Target Retrieval Solution (K800421-2, Aligent, Santa Clara, CA, USA) with high pH at 97°. Tryptase epitope retrieval was performed for 20 min in PBS + 0.05% Pronase XIV enzyme at 37°. For all immunostainings, sections were further processed using a fully automated immunostainer (Lab Vision Autostainer 360; Thermo Fisher Scientific, Waltham, MA, USA) and the EnVision FLEX visualization system (K8000, Agilent, Santa Clara, CA, USA, RRID:AB_2890017), according to manufacturer’s instruction. In short: sections were rinsed in EnVision FLEX Wash Buffer (K800721-2, Aligent) for 5 min, followed by 5 min in Peroxidase-Blocking Reagent and a 5 min rinse in EnVision FLEX Wash Buffer. Sections were incubated with the aforementioned primary antibody for 60 min. After incubation, sections were rinsed for 10 min in EnVision FLEX Wash Buffer and incubated for 15 min with EnVision FLEX Mouse (LINKER) (K802121-2, Agilent). Again, sections were rinsed for 10 min in EnVision FLEX Wash Buffer and incubated with EnVision FLEX HRP Solution (Agilent) for 30 min, whereafter another 10 min rinse in EnVision FLEX Wash Buffer was performed. Sections were incubated with a mixture of EnVision FLEX 3,3′-diaminobenzidine (DAB) + and Substrate Solution (Agilent) for 10 min and rinsed in tap water for 10 min. Sections were counterstained using hematoxylin before dehydration in ethanol and xylene and cover slipping.

### 4.9. Zonula Occludens-1 Immunofluorescence 

Sections were deparaffinized in xylene, rinsed in ethanol and demi water, followed by heat-induced antigen retrieval with Tris-EDTA buffer solution (pH 9.0) at 95 °C for 10 min. The sections were rinsed in Tris-buffered Saline (TBS). Nonspecific binding was prevented with TBS-BT (0.1% BSA and 0.3% Triton, 30 min). Subsequently, the sections were incubated with primary antibody (Rabbit anti-TJP1/ZO-1, 1:200, Abcam, Cambridge, UK, ab221547, RRID: AB 2892660) in TBS-BT for 1 h at room temperature. Then, sections were rinsed in TBS and the secondary antibody (Anti-rabbit-Alexa Fluor Plus 488, 1:200, Thermo Scientific, Waltham, MA, USA, A32790, RRID: AB 2762833) in TBS-BT was applied for 1 h at room temperature. This was followed by another rinse with TBS and DAPI (1:1000) in TBS incubation for 10 min. Lastly, the slides were rinsed with TBS and cover-slipped with Fluorsave (Merck Millipore, 345789, Burlington, MA, USA). 

### 4.10. Post-Processing of Sections

The CD8, CD68 and tryptase stained sections were digitized on a Pannoramic 1000 slide scanner (3DHISTECH Ltd., Budapest, Hungary) employing a 20× magnifying objective. All high-resolution digital images (0.25 µm/pixel) were both visualized and exported to tag image file format (TIFF) (1:4 scale, 8-bit, jpeg with 80% compression) using CaseViewer software (version 2.4; 3DHISTECH Ltd., Budapest, Hungary). Images of ZO-1 were captured (as TIFF files) using an immunofluorescent microscope (Axio Imager A2, Carl Zeiss, Oberkochen, Germany) with a 20× magnifying objective. 

For every specimen, CD68, CD8 and tryptase stainings were co-registered to the corresponding HE scan through a custom written intensity-based automatic image registration MATLAB script (MATLAB R2020a; MathWorks Inc., Natick, MA, USA) [43]. When the automated registration script failed to accurately register the other immunohistochemical stainings to HE, manual registration based on landmark selection in the HE reference section and target-stained section was performed. For every subject, a region of interest (ROI) (with 10 intact villi) was defined. The chosen ROI was selected from the realigned images with the polygon function in MATLAB [44]. These detailed crop images were computationally segmented in ImageJ (version 1.47v, National Institute of Health, Bethesda, MD, USA) by making use of the color deconvolution tool [45]. A threshold was set to eliminate background noise, whereafter double-blinded manual selection of the 10 individual villi was performed. Finally, ImageJ measured the number of positive stained immune cells present within every villus, which eventually was converted to the number of positive stained cells per mm^2^. For ZO-1, TIFF files were loaded in ImageJ, whereafter a background correction was performed and signal of non-specific binding was eliminated. A threshold was set to eliminate background noise and the 10 villi of interest were double-blinded manually selected, creating an intensity score of ZO-1 (Mean Gray Value (MGV) range, 0–255; higher score indicates higher intestinal integrity).

To determine whether microbiota, its metabolites and cytokines were related to intestinal inflammation and integrity, the cohort was divided into a high and low intestinal inflammation group as well as a high and low intestinal integrity group. First, a normalized total inflammation score was calculated for every subject. Therefore, the cell count of the immune cells was normalized for every subject using the following formula: Normalized immune cell score = (number of cells/mm^2^)/(group average). Finally, the normalized scores for cytotoxic T cells, macrophages and mast cells were added together to form the normalized total inflammation score. Based on the mean of the total inflammation score, the group was divided into low and high intestinal inflammation. Moreover, the group was divided into low and high intestinal integrity based on the mean of the ZO-1 score.

### 4.11. Statistics 

Statistical analyses were performed using R version 4.2.2. Data visualizations were generated using ‘ggplot2’ [46]. All statistical tests were two-tailed, and *p*-values less than 0.05 were considered statistically significant. 

Microbiota data processing utilized the ‘phyloseq’ package [47], filtering taxa based on prevalence and relative abundance using the method described elsewhere [48]. Alpha diversity (Shannon diversity index) was calculated using the ‘diversity’ function from the ‘vegan’ package. The between-group difference in beta-diversity was tested using permutational multivariate analysis of variance (PERMANOVA) with the ‘adonis2’ function from the ‘vegan’ package (v2.6-6.1; CRAN—Package vegan (r-project.org)). The differential abundance of microbial taxa was determined using DESeq2 [49]. The significance of differences between groups was evaluated by applying the Wald test to the fitted models. *p*-values were adjusted for multiple comparisons using the Benjamini–Hochberg procedure. Lastly, we included sex as a covariate when analyzing differences in metabolites.

For the histological data, including immune cell counts and ZO-1 intensity, group comparisons were made using linear models. Data transformation (Box-Cox transformation) was applied where necessary to meet model assumptions. Outliers were identified and removed using the interquartile range rule on model residuals.

## Figures and Tables

**Figure 1 pharmaceuticals-17-00918-f001:**
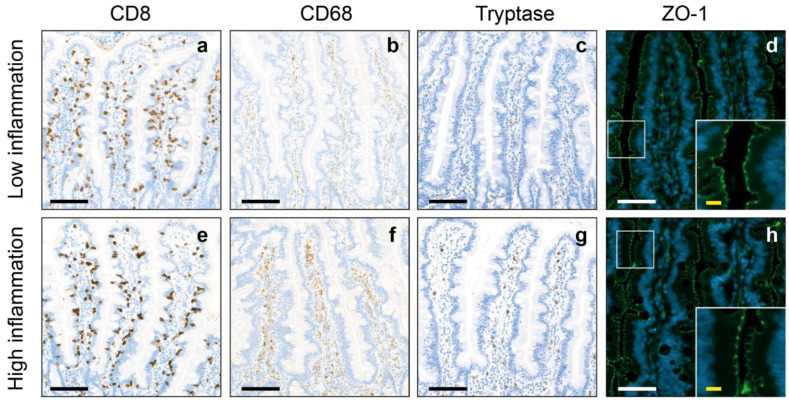
Representative images of the high and low intestinal inflammation groups. Representative images of cytotoxic T cells (CD8), macrophages (CD68) and mast cells (Tryptase) in the jejunum of individuals corresponding to the low inflammation group (**a**–**d**) and individuals corresponding to the high inflammation group (**e**–**h**). (**d**,**h**) Representative images of Zonula occludens-1 (ZO-1) in the jejunum of individuals with high (**d**) and low intestinal inflammation (**h**). The white boxes indicate regions of interest placed in the lower corner of (**d**,**h**). Black/white scale bar = 100 µm, yellow scale bar = 40 µm).

**Figure 2 pharmaceuticals-17-00918-f002:**
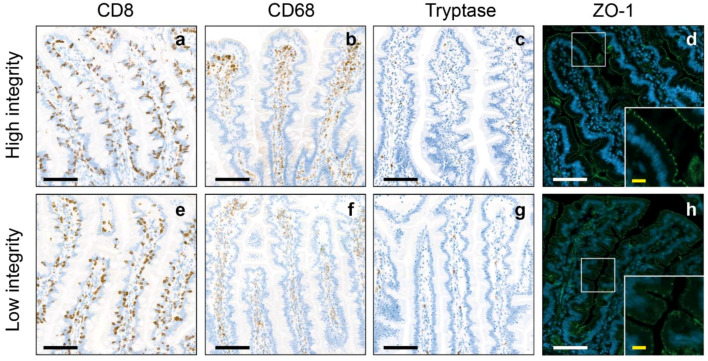
Representative images of the high and low intestinal integrity groups. Representative images of cytotoxic T cells (CD8), macrophages (CD68) and mast cells (Tryptase) in the jejunum of individuals corresponding to the low integrity group (**a**–**d**) and individuals corresponding to the high integrity group (**e**–**h**). (**d**,**h**) Representative images of Zonula occludens-1 (ZO-1) in the jejunum of individuals with a high (**d**) and low intestinal integrity (**h**). The white boxes indicate regions of interest placed in the lower corner of (**d**,**h**). Black/white scale bar = 100 µm, yellow scale bar = 40 µm).

**Figure 3 pharmaceuticals-17-00918-f003:**
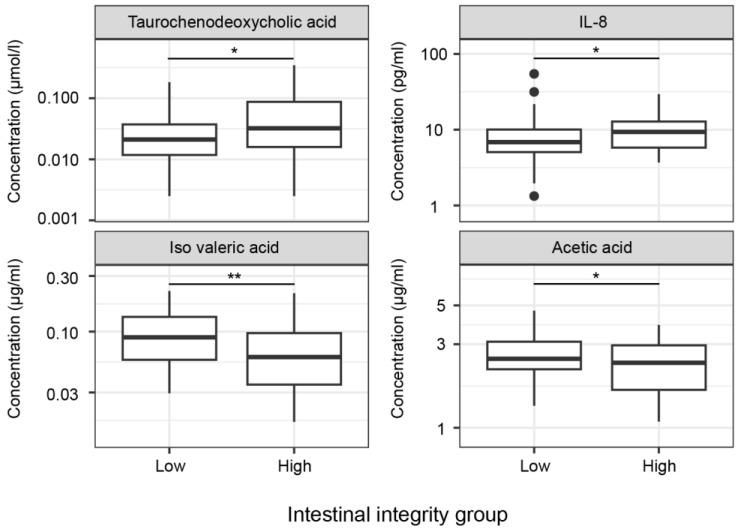
Significant differences in metabolites and cytokines between the low and high intestinal integrity groups. Levels of tauro-chenodeoxycholic acid (*p* = 0.019) and IL-8 (*p* = 0.011) were significantly higher, while levels of iso valeric acid (*p* = 0.003) and acetic acid (*p* = 0.043) were significantly lower in the high intestinal integrity group. Abbreviations: IL, interleukin, * = *p* < 0.05, ** = *p* < 0.01.

**Figure 4 pharmaceuticals-17-00918-f004:**
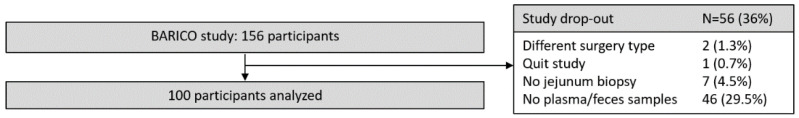
Flowchart with included participants. Out of the 156 participants enrolled in the BARICO study, 2 participants were excluded due to a last-minute change of surgery (sleeve gastrectomy); 1 other participant dropped out of the study before the surgery; from 7 participants, no jejunum biopsy was collected; and from 46 participants, no plasma and/or fecal samples were available, leaving 100 participants eligible for analysis.

**Table 1 pharmaceuticals-17-00918-t001:** Characteristics of participants (*n* = 100).

		Intestinal Inflammation			Intestinal Integrity		
	All (*n* = 100)	Low (*n* = 52)	High (*n* = 48)	*p*-Value	Low (*n* = 62)	High (*n* = 38)	*p*-Value
**Age, mean ± SD, y**	46.48 ± 5.73	46.48 ± 5.68	46.47 ± 5.85	0.993	46.74 ± 5.32	46.05 ± 6.40	0.564
**Sex, women, n (%)**	82 (82.0)	43 (82.69)	39 (81.25)	0.528	46 (74.19)	36 (94.74)	0.007
**Body length, mean ± SD, m**	1.71 ± 0.07	1.71 ± 0.08	1.71 ± 0.07	0.997	1.73 ± 0.08	1.69 ± 0.06	0.013
**Body weight, mean ± SD, kg**	122.69 ± 15.69	120.96 ± 16.08	124.56 ± 15.20	0.253	124.21 ± 17.0	120.19 ± 13.20	0.216
**BMI, mean ± SD, kg/m^2^**	41.61 ± 3.92	40.98 ± 3.60	42.28 ± 4.17	0.097	41.39 ± 3.63	41.97 ± 4.37	0.476
**WC ^a^, mean ± SD, cm**	124.75 ± 11.14	124.16 ± 11.27	125.36 ± 11.11	0.613	125.03 ± 11.57	124.34 ± 10.64	0.777
**Comorbidities, n (%)**							
Pre-diabetes	23 (23.0)	12 (23.08)	11 (22.92)	0.587	16 (25.81)	7 (18.42)	0.275
Diabetes	14 (14.0)	5 (9.62)	9 (18.75)	0.152	10 (16.13)	4 (10.53)	0.319
Hypertension	71 (71.0)	37 (71.15)	34 (70.83)	0.573	45 (72.58)	26 (68.42)	0.411
Irritable bowel syndrome	5 (5.0)	3 (5.77)	2 (4.17)	0.538	4 (6.45)	1 (2.63)	0.368
Ulcerative colitis	1 (1.0)	1 (1.92)	0 (0)	0.520	1 (1.61)	0 (0)	0.620
**Smoking current, n (%)**	6 (6.0)	3 (5.77)	3 (6.25)	0.622	3 (4.84)	3 (7.89)	0.413
**Consuming alcohol, n (%)**	40 (40.0)	18 (34.62)	22 (45.83)	0.174	25 (40.32)	15 (39.47)	0.551
Alcohol consumption, median (IQR), units per week	1.0 (3.0)	1.0 (3.25)	1.5 (3.0)	0.965	2.0 (4.0)	1.0 (2.0)	0.188
**Blood pressure, mean ± SD**							
Systolic (mm Hg)	136.72 ± 16.30	136.17 ± 18.06	137.31 ± 14.31	0.729	136.90 ± 16.49	136.42 ± 16.21	0.887
Diastolic (mm Hg)	84.78 ± 7.96	81.29 ± 8.03	85.42 ± 7.92	0.445	84.84 ± 8.25	84.68 ± 7.57	0.926
**Gut pathology, mean ± SD**							
Cytotoxic T cells, number/mm^2^	1040.16 ± 285.17	1021.97 ± 269.94	1059.87 ± 302.42	0.509	1007.72 ± 297.37	1093.10 ± 259.15	0.147
Macrophages, number/mm^2^	628.36 ± 308.72	482.25 ± 236.27	786.64 ± 298.28	<0.001	555.63 ± 268.25	747.02 ± 331.30	0.002
Mast cells, number/mm^2^	103.18 ± 66.19	58.05 ± 40.22	152.08 ± 52.68	<0.001	104.12 ± 72.00	101.65 ± 56.30	0.857
ZO-1 intensity (MGV)	76.34 ± 6.48	75.55 ± 6.81	77.19 ± 6.05	0.205	72.43 ± 2.57	82.72 ± 5.83	<0.001

^a^ Complete data were available for 89 participants. Abbreviations: SD, standard deviation; BMI, body mass index; WC, waist circumference; ZO-1, zonula occludens-1; MGV, mean gray value.

## Data Availability

De-identified data from the BARICO study will be made available upon request after approval by the study investigators.

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
