# Peer review of "Impact of Microbiota and Metabolites on Intestinal Integrity and Inflammation in Severe Obesity"

_pharmaceuticals, 2024, doi:10.3390/ph17070918_

Round 1
Reviewer 1 Report
Comments and Suggestions for Authors
This manuscript analyzed the associations between the intestinal integrity/inflammation, fecal microbiota and plasma microbiota-derived metabolites in severely obese adults. This study is well organized, and the results are interesting, but there are several major concerns:
1. The introduction of the manuscript introduced the role of microbiota and metabolites in intestinal inflammation but didn’t provide any information about their roles in the intestinal barrier functions/dysfunctions or the intestinal integrity. This information is necessary to help readers to understand the study results better.
2. In the method, it is necessary to specific the criteria used to divide the patients into low and high groups in intestinal integrity based on the mean of the ZO-1 score or intestinal inflammation based on the total inflammation score since microbiota and the metabolites also exist in normal intestinal tissues.
3. In the Table 1 Characteristics of participants, it is also necessary to specific if any participants have diseases related to GI tract.
4. It is a major defect in this study that there is no normal control groups. I would suggest the authors to find the reported data of normal control groups in publications and compare the collected data to the reported ones, which might help the interpretation of the data.
5. There are a great number of grammar mistakes in the manuscript.
Comments on the Quality of English LanguageThere are a great number of grammar mistakes in the manuscript. Some sentences are confusing. The manuscript requires extensive English editing.
Author Response
|
1. Summary |
|
|
|
Thank you very much for taking the time to review this manuscript. Please find the comments and suggestions of reviewers in black, our detailed responses below in blue and the corresponding revisions/corrections in blue italic and in track changes in the submitted files. |
||
|
2. Point-by-point response to Comments and Suggestions for Authors |
||
|
Comments 1: The introduction of the manuscript introduced the role of microbiota and metabolites in intestinal inflammation but didn’t provide any information about their roles in the intestinal barrier functions/dysfunctions or the intestinal integrity. This information is necessary to help readers to understand the study results better. |
||
|
Response 1: Thank you for pointing this out. We agree with this comment. Therefore we have included some additional information about gut bacteria and their link with intestinal integrity in the introduction. (Page 2, line 52 to 58) “As intestinal inflammation is associated with reduced intestinal barrier function[1-3], the gut microbiota could induce intestinal permeability through intestinal inflammation. Moreover, in an in vitro model of intestinal epithelium barrier monolayer it was demonstrated that a combination of several gut bacteria, including Lactobacillus Rhamnosus, Bifidobacterium Lactis and Bifidobacterium Longum increased the expression of tight junction proteins, including zonulin-1 and -2, occludin and claudin-1, indicating a role of the microbiota in intestinal integrity[4].” |
||
|
Comments 2: In the method, it is necessary to specific the criteria used to divide the patients into low and high groups in intestinal integrity based on the mean of the ZO-1 score or intestinal inflammation based on the total inflammation score since microbiota and the metabolites also exist in normal intestinal tissues. |
||
|
Response 2: We agree that it is important to specify how we make the distinction between high and low intestinal inflammation/integrity. This has already been done in the method section. To introduce this a bit more we included one extra sentence. (Page 12, line 457 to 467) “To determine whether microbiota, its metabolites and cytokines were related to intestinal inflammation and integrity the cohort was divided into a high and low intestinal inflammation group as well as a high and low intestinal integrity group. First, a normalized total inflammation score was calculated for every subject. Therefore, the cell count of the immune cells was normalized for every subject using the following formula: Normalized immune cell score = (number of cells/ mm2)/(group average). Finally, the normalized scores for cytotoxic T-cells, macrophages and mast cells were added together to form the normalized total inflammation score. Based on the mean of the total inflammation score, the group was divided into low and high intestinal inflammation. Moreover, the group was divided into low and high intestinal integrity based on the mean of the ZO-1 score.” Comments 3: In the Table 1 Characteristics of participants, it is also necessary to specific if any participants have diseases related to GI tract. |
||
|
Response 3: Thank you for this suggesting. We agree with the reviewer and have included the prevalence of irritable bowel syndrome and ulcerative colitis in our cohort in Table 1. Only 5 patients had irritable bowel syndrome and 1 patient had ulcerative colitis. As the prevalence is low, we did not found significant differences in the prevalence of these diseases between the high and low intestinal inflammation and integrity groups. (Page 3) (see attachment for table 1)
Comments 4: It is a major defect in this study that there is no normal control groups. I would suggest the authors to find the reported data of normal control groups in publications and compare the collected data to the reported ones, which might help the interpretation of the data. Response 4: We agree with the reviewer that it is important to include a control group. However, as this is a sub-study of the BARICO study, which only included patients with obesity eligible for bariatric surgery, we did not have such group. However, as suggested by the reviewer we compared our data to immunohistochemical data of healthy lean controls from different studies. Moreover, we compared the metabolite and cytokine data with normal ranges of these metabolites/cytokines in healthy persons. Nonetheless, some studies used different regions of the intestine and methods of analysis may differ. Therefore, in future studies it is important to include a control group. (Page 6 to 7, line 182 to 197) “Monteiro-Sepulveda and co-workers demonstrated that classical cytotoxic T-cells and macrophages have a higher abundancy in individuals with obesity compared to lean individuals[5], whereas mast cell density was similar in obesity and lean controls. More specifically, they found 1550 cytotoxic T-cells/mm2, 150 macrophages/mm2 and 100 mast cells/mm2 in patients with obesity, compared to 1000 cytotoxic T-cells/ mm2, 100 macrophages/mm2and 100 mast cells/ mm2 in lean controls. In our study we found a similar cytotoxic T-cell count(1040 cells/mm2) compared to the lean control group in the study of Monteiro-Sepulveda and colleagues(1000 cells/mm2). However, the macrophage cell density (628 cells/mm2) was higher in our study, compared to the cohort with obesity (150 cells/mm2) and lean controls (100 cells/mm2) in the study of Monteiro-Sepulveda et al. Another study showed approximately 900 cytotoxic T-cells/mm2 and approximately 1100 macrophages/mm2 in the lamina propria of the colon from patients with Crohn’s disease compared to 200 cytotoxic T-cells/mm2 and 550 macrophages/mm2 in healthy controls[6]. In addition, it has been found that macrophages were increased in the lamina propria of the duodenum in patients with Crohn’s disease (110 cells/mm2) compared to controls (50 cell/mm2)[7].” (Page 7 to 8, line 240 to 257) “We detected higher IL-8 and TCDC plasma levels, as well as lower levels of iso valeric acid and acetic acid in individuals with a high intestinal integrity. IL-8 is known to be a chemoattractant of neutrophils which form an important line of defense against bacterial pathogens[8], suggesting that higher levels of IL-8 could maintain a higher intestinal integrity. However, the mean concentration of IL-8 in both the low and high permeability group are still within normal range (<62 pg/ml)[9], which may suggest that pathology may be limited in these groups. In an organoid-derived epithelial monolayer culture from a patient with ulcerative colitis it was shown that acetate stimulation could prevent alterations of the monolayer integrity upon inflammation, suggesting that acetate has barrier protective properties[10]. Moreover, in human colonic Caco-2 cells it was shown that acetate maintains gut integrity by increasing cell survival[11]. In our study however, the high intestinal integrity group showed lower acetic acid levels. Therefore, we compared the acetic acid concentration of our cohort to the normal range of acetic acid in healthy people (3.6ug/ml)[12], and found little differences. Furthermore, iso valeric acid and TCDC are negatively associated with Crohn’s disease and inflammation [13-15], while in our high intestinal integrity group these levels where higher compared to the low intestinal integrity group. Nonetheless, the mean TCDC concentration of both the low and high inflammation group are similar to those of healthy individuals (0.057µmol/L)[16].” Comments 5: There are a great number of grammar mistakes in the manuscript. Response 5: Thanks for the critical note. We revised the manuscript and improved grammar and typos.
See attachment for references. |
||

Reviewer 2 Report
Comments and Suggestions for Authors
Some major points must be taken into account:
l It is unacceptable not to have a healthy group whose results can be used in the comparison and to rely on comparison with results from other studies.
l The authors indicated no differences in the gut microbiota. It is not clear in the results which strains were identified, and it is necessary to compare them with healthy people.
l It is necessary to distinguish between biochemical parameters such as cytokines and metabolites of the microbiota, and it is not appropriate to combine and mix them.
l What I understood is that the study was conducted on 100 participants who were divided into 4 groups as mentioned in the abstract, while the numbers in Table 1 are confusing. Please check.
l The basic concept in the study is the effect of the microbiota and metabolites. The presence of the results for these concepts in the supplementary file and not showing them in the manuscript weakens the manuscript.
l Making a comparison with the results of a healthy group will give the results a better appearance and provide the opportunity for a broader and more accurate discussion.
Author Response
|
1. Summary |
|
|
|
We thank the reviewer for the comments and suggestions on our manuscript. We appreciate all comments and have responded as truthfully as possible. Please find the comments and suggestions of reviewers in black, our detailed responses below in blue and the corresponding revisions/corrections in blue italic and in track changes in the re-submitted files. 2. Point-by-point response to comments and suggestions for authors Comments 1: It is unacceptable not to have a healthy group whose results can be used in the comparison and to rely on comparison with results from other studies. Response 1: We agree with the reviewer that it is very important to include a control group in this study. However, as this is a sub-study of the BARICO study, which only included patients with obesity eligible for bariatric surgery, we did not have such group. However, as suggested by the reviewer we compared our data to immunohistochemical data of healthy lean controls from different studies. Moreover, we compared the metabolite and cytokine data with normal ranges of these metabolites/cytokines in healthy persons. Nonetheless, some studies used different regions of the intestine and methods of analysis may differ. Therefore, in future studies it is important to include a control group. (Page 6 to 8, line 182 to 197) “Monteiro-Sepulveda and co-workers demonstrated that classical cytotoxic T-cells and macrophages have a higher abundancy in individuals with obesity compared to lean individuals[1], whereas mast cell density was similar in obesity and lean controls. More specifically, they found 1550 cytotoxic T-cells/mm2, 150 macrophages/mm2 and 100 mast cells/mm2 in patients with obesity, compared to 1000 cytotoxic T-cells/ mm2, 100 macrophages/mm2and 100 mast cells/ mm2 in lean controls. In our study we found a similar cytotoxic T-cell count(1040 cells/mm2) compared to the lean control group in the study of Monteiro-Sepulveda and colleagues(1000 cells/mm2). However, the macrophage cell density (628 cells/mm2) was higher in our study, compared to the cohort with obesity (150 cells/mm2) and lean controls (100 cells/mm2) in the study of Monteiro-Sepulveda et al. Another study showed approximately 900 cytotoxic T-cells/mm2 and approximately 1100 macrophages/mm2 in the lamina propria of the colon from patients with Crohn’s disease compared to 200 cytotoxic T-cells/mm2 and 550 macrophages/mm2 in healthy controls[2]. In addition, it has been found that macrophages were increased in the lamina propria of the duodenum in patients with Crohn’s disease (110 cells/mm2) compared to controls (50 cell/mm2)[3].” (Page 7 to 8, line 240 to 257) “We detected higher IL-8 and TCDC plasma levels, as well as lower levels of iso valeric acid and acetic acid in individuals with a high intestinal integrity. IL-8 is known to be a chemoattractant of neutrophils which form an important line of defense against bacterial pathogens[4], suggesting that higher levels of IL-8 could maintain a higher intestinal integrity. However, the mean concentration of IL-8 in both the low and high permeability group are still within normal range (<62 pg/ml)[5], which may suggest that pathology may be limited in these groups. In an organoid-derived epithelial monolayer culture from a patient with ulcerative colitis it was shown that acetate stimulation could prevent alterations of the monolayer integrity upon inflammation, suggesting that acetate has barrier protective properties[6]. Moreover, in human colonic Caco-2 cells it was shown that acetate maintains gut integrity by increasing cell survival[7]. In our study however, the high intestinal integrity group showed lower acetic acid levels. Therefore, we compared the acetic acid concentration of our cohort to the normal range of acetic acid in healthy people (3.6ug/ml)[8], and found little differences. Furthermore, iso valeric acid and TCDC are negatively associated with Crohn’s disease and inflammation [9-11], while in our high intestinal integrity group these levels where higher compared to the low intestinal integrity group. Nonetheless, the mean TCDC concentration of both the low and high inflammation group are similar to those of healthy individuals (0.057µmol/L)[12].” Comments 2: The authors indicated no differences in the gut microbiota. It is not clear in the results which strains were identified, and it is necessary to compare them with healthy people. Response 2: Thanks for this comment. With 16S rDNA amplicon sequencing the microbiota alpha and beta diversity was determined. All detectable strains were included in this research. As the list of all bacteria is quite large (8 pages), supplementary table 4 and 6 visualize the included bacteria and its difference between the high and low intestinal inflammation and intestinal integrity groups. It would be very interesting to compare the microbiota of our cohort with a healthy control group. However, as already mentioned in Response 1, we did not include such group in our study. Moreover, it is difficult to compare our microbiota data to previous literature in healthy controls as many factors can influence the microbiota also in healthy individuals. Therefore, we did not include such comparisons in our manuscript. Nonetheless, we strongly agree that in future research a healthy control group should be included to interpret results in a more distinct manner. Comments 3: It is necessary to distinguish between biochemical parameters such as cytokines and metabolites of the microbiota, and it is not appropriate to combine and mix them. Response 3: Thanks for this comment. To distinguish between biochemical parameters and metabolites of the microbiota we made two sub-headings; one for the metabolites and one for the cytokines (page 5 to 6, line 139 to 164). Moreover, we corrected this throughout the entire manuscript. “2.4. Metabolites To assess differences in metabolites between the intestinal inflammation and integrity groups, sex was included as a covariate. No significant differences were found in metabolite concentrations between intestinal inflammation groups (Table S3). The significantly different metabolites between the high and low intestinal integrity groups are depicted in Figure 3 and Table S5. Levels of tauro-chenodeoxycholic acid (TCDC) (p=0.019) were significantly higher, whereas levels of iso valeric acid (p=0.003) and acetic acid (p=0.043) were significantly lower in the high intestinal integrity group. No significant differences were found for cholic acid (CA), chenodeoxycholic acid (CDC), deoxycholic acid (DCA), glycocholic acid (GCA), glycochenodeoxycholic acid (GCDC), glycodeoxycholic acid (GDC), glycolitocholic acid-3-sulphate (GLC-3S), tauro-deoxycholic acid (TDC), ursodeoxycholic acid (UDC), propionic acid, butyric acid, iso butyric acid and methyl butyric acid, between the high and low intestinal integrity group (Table S5).
2.5 Cytokines Sex, was included as a covariate to study cytokines concentrations in the intestinal inflammation and integrity groups. No significant differences were found in cytokine concentrations between intestinal inflammation groups (Table S3). The significantly different metabolites between the high and low intestinal integrity groups are depicted in Figure 3 and Table S5. Levels of IL-8 (p=0.011) were significantly higher in the high intestinal integrity group compared to the low intestinal integrity group. No significant differences were found for high sensitive (hs) C-reactive protein (CRP), serum amyloid A (SAA), haptoglobin, LPS binding protein (LBP), tumor necrosis factor alpha (TNFα), interleukin-1β (IL-1β), IL-6, IL-4 and IL-10 between the high and low intestinal integrity group (Table S5).”
Comments 4: What I understood is that the study was conducted on 100 participants who were divided into 4 groups as mentioned in the abstract, while the numbers in Table 1 are confusing. Please check. Response 4: We agree with the reviewer that the numbers in Table 1 can be confusing. We divided our cohort in two groups based on the mean of the normalized total immune cell score. Additionally, we divided our cohort in two groups based on the mean of the zonula occludens-1 intensity score. The normalized total immune cell score was normally distributed and therefore lead to an equal distribution with 52 patients in the low intestinal inflammation group and 48 patients in the high intestinal inflammation group. The zonula occludens-1 intensity score however, had a right-skewed distribution, with more low than high values. This explains why more patients were included in the low intestinal integrity group (n=62) compared to the high intestinal integrity groups (n=38). Comments 5: The basic concept in the study is the effect of the microbiota and metabolites. The presence of the results for these concepts in the supplementary file and not showing them in the manuscript weakens the manuscript. Response 5: We understand that the reviewer recommends to include the supplementary tables in the manuscript. However, since only a few results were significant we chose to only include those in the manuscript. It may be distracting when all microbiota, metabolite and cytokine data will be included in the manuscript (as this concerns 6 tables over 24 pages). As in this case, the reader needs to search through many tables to find the significant results. Comments 6: Making a comparison with the results of a healthy group will give the results a better appearance and provide the opportunity for a broader and more accurate discussion. Response 6: We definitely agree with the reviewer that having a healthy group would have provided a more accurate discussion Therefore, in our discussion we tried to compensate for the lack of a control group and compared our immunohistochemical data with previous literature about intestinal inflammation in bowel disorders but also in healthy controls. Moreover, we compared the metabolite and cytokine concentrations with normal ranges of healthy individuals. With this information we tried to better interpret our results. Nonetheless, in future studies a control group should be included. (Page 6 to 8, line 182 to 197) “Monteiro-Sepulveda and co-workers demonstrated that classical cytotoxic T-cells and macrophages have a higher abundancy in individuals with obesity compared to lean individuals[1], whereas mast cell density was similar in obesity and lean controls. More specifically, they found 1550 cytotoxic T-cells/mm2, 150 macrophages/mm2 and 100 mast cells/mm2 in patients with obesity, compared to 1000 cytotoxic T-cells/ mm2, 100 macrophages/mm2and 100 mast cells/ mm2 in lean controls. In our study we found a similar cytotoxic T-cell count(1040 cells/mm2) compared to the lean control group in the study of Monteiro-Sepulveda and colleagues(1000 cells/mm2). However, the macrophage cell density (628 cells/mm2) was higher in our study, compared to the cohort with obesity (150 cells/mm2) and lean controls (100 cells/mm2) in the study of Monteiro-Sepulveda et al. Another study showed approximately 900 cytotoxic T-cells/mm2 and approximately 1100 macrophages/mm2 in the lamina propria of the colon from patients with Crohn’s disease compared to 200 cytotoxic T-cells/mm2 and 550 macrophages/mm2 in healthy controls[2]. In addition, it has been found that macrophages were increased in the lamina propria of the duodenum in patients with Crohn’s disease (110 cells/mm2) compared to controls (50 cell/mm2)[3].” (Page 7 to 8, line 240 to 257) “We detected higher IL-8 and TCDC plasma levels, as well as lower levels of iso valeric acid and acetic acid in individuals with a high intestinal integrity. IL-8 is known to be a chemoattractant of neutrophils which form an important line of defense against bacterial pathogens[4], suggesting that higher levels of IL-8 could maintain a higher intestinal integrity. However, the mean concentration of IL-8 in both the low and high permeability group are still within normal range (<62 pg/ml)[5], which may suggest that pathology may be limited in these groups. In an organoid-derived epithelial monolayer culture from a patient with ulcerative colitis it was shown that acetate stimulation could prevent alterations of the monolayer integrity upon inflammation, suggesting that acetate has barrier protective properties[6]. Moreover, in human colonic Caco-2 cells it was shown that acetate maintains gut integrity by increasing cell survival[7]. In our study however, the high intestinal integrity group showed lower acetic acid levels. Therefore, we compared the acetic acid concentration of our cohort to the normal range of acetic acid in healthy people (3.6ug/ml)[8], and found little differences. Furthermore, iso valeric acid and TCDC are negatively associated with Crohn’s disease and inflammation [9-11], while in our high intestinal integrity group these levels where higher compared to the low intestinal integrity group. Nonetheless, the mean TCDC concentration of both the low and high inflammation group are similar to those of healthy individuals (0.057µmol/L)[12].”
For references see the attachment |
||

Round 2
Reviewer 1 Report
Comments and Suggestions for Authors
The revision of the manuscript is well done.
Reviewer 2 Report
Comments and Suggestions for Authors
The suggested modifications were carried out.